# Targeted Double-Stranded cDNA Sequencing-Based Phase Analysis to Identify Compound Heterozygous Mutations and Differential Allelic Expression

**DOI:** 10.3390/biology10040256

**Published:** 2021-03-24

**Authors:** Hiroki Ura, Sumihito Togi, Yo Niida

**Affiliations:** 1Center for Clinical Genomics, Kanazawa Medical University Hospital, 1-1 Daigaku, Uchinada, Kahoku, Ishikawa 920-0923, Japan; togi@kanazawa-med.ac.jp (S.T.); niida@kanazawa-med.ac.jp (Y.N.); 2Division of Genomic Medicine, Department of Advanced Medicine, Medical Research Institute, Kanazawa Medical University, 1-1 Daigaku, Uchinada, Kahoku, Ishikawa 920-0923, Japan

**Keywords:** phase analysis, compound heterozygous mutation, next-generation sequencing, targeted double-stranded cDNA sequencing, allelic expression

## Abstract

**Simple Summary:**

Phase analysis to distinguish between *in cis* and *in trans* heterozygous mutations is important for clinical diagnosis because *in trans* compound heterozygous mutations cause autosomal recessive diseases. However, conventional phase analysis is limited because of the large target size of genomic DNA. Here, we performed a targeted double-stranded cDNA sequencing-based phase analysis to resolve the limitation of distance using direct adapter ligation library preparation and paired-end sequencing; we elucidated that two heterozygous mutations on a patient with Wilson disease are *in trans* compound heterozygous mutations. Furthermore, we detected the differential allelic expression. Our results indicate that a targeted double-stranded cDNA sequencing-based phase analysis is useful for determining compound heterozygous mutations and confers information on allelic expression.

**Abstract:**

There are two combinations of heterozygous mutation, i.e., *in trans,* which carries mutations on different alleles, and *in cis,* which carries mutations on the same allele. Because only *in trans* compound heterozygous mutations have been implicated in autosomal recessive diseases, it is important to distinguish them for clinical diagnosis. However, conventional phase analysis is limited because of the large target size of genomic DNA. Here, we performed a genetic analysis on a patient with Wilson disease, and we detected two heterozygous mutations chr13:51958362;G>GG (NM_000053.4:c.2304dup r.2304dup p.Met769HisfsTer26) and chr13:51964900;C>T (NM_000053.4:c.1841G>A r.1841g>a p.Gly614Asp) in the causative gene *ATP7B*. The distance between the two mutations was 6.5 kb in genomic DNA but 464 bp in mRNA. Targeted double-stranded cDNA sequencing-based phase analysis was performed using direct adapter ligation library preparation and paired-end sequencing, and we elucidated they are *in trans* compound heterozygous mutations. Trio analysis showed that the mutation (chr13:51964900;C>T) derived from the father and the other mutation from the mother, validating that the mutations are *in trans* composition. Furthermore, targeted double-stranded cDNA sequencing-based phase analysis detected the differential allelic expression, suggesting that the mutation (chr13:51958362;G>GG) caused downregulation of expression by nonsense-mediated mRNA decay. Our results indicate that targeted double-stranded cDNA sequencing-based phase analysis is useful for determining compound heterozygous mutations and confers information on allelic expression.

## 1. Introduction

Next-generation sequencing (NGS) is a powerful technology used in the clinical field for genetic diagnosis [1,2,3]. At present, the use of NGS in clinical diagnosis is largely for comprehensive analysis, such as whole-genome sequencing, whole-exome sequencing, and gene targeting panel sequencing. Phase analysis, which detects specific compound heterozygous mutations, is not commonly performed using NGS technology.

During genetic diagnosis, multiple heterozygous mutations could be detected at specific loci. There are two combinations of heterozygous mutations. *In cis* heterozygous loss-of-function mutation still retains one functionally active allele as both mutations are located at the same allele (Figure 1A), whereas *in trans* compound heterozygous mutation does not retain any functionally active alleles as each mutation is found at a different allele. A patient who has *in trans* compound heterozygous loss-of-function mutation will be affected by an autosomal recessive disease similar to a patient who has loss-of-function homozygous mutation. Therefore, the clinical diagnoses must distinguish between *in trans* and *in cis* heterozygous mutations when there are more than two heterozygous mutations at a particular gene locus.

However, a conventional phase analysis uses Sanger sequencing, which is limited by the distance between the two mutations and the required number of sequencing reads. The clone into the plasmid or fosmid must include the sequence of both mutations. Recently, an alternative approach for genetic diagnosis has been available with NGS technology of short read DNA sequencing (DNA-seq) [4,5,6]; however, it is also limited by the relatively short distances spanned by the reads. It is difficult to use except in cases of two heterozygous mutations in the same exon. To resolve the distance limitation, we used a targeted double-stranded cDNA sequencing-based phase analysis to detect two mutations in a single or paired-end read by removing large intron sequences by splicing (Figure 1B). In traditional RNA-seq, RNA is fragmented before cDNA synthesis [7]. To save the read including both mutations, we used the SMARTer RNA-seq method to first synthesize the full-length double-stranded cDNA once, and then fragment the double-stranded cDNA for NGS. Furthermore, the specific *ATP7B* double-stranded cDNA spanning two mutations was amplified from SMARTer cDNA that was, then, directly ligated to the adapter without fragmentation to ensure all paired-end reads contained both mutations.

Allele-specific expression of two alleles in a diploid individual may be potentially imbalanced, thereby contributing to phenotypic variation and disease pathophysiology among individuals [8,9]. Nonsense or frameshift mutations that induce nonsense-mediated mRNA decay (NMD) strongly affect imbalanced allelic expression due to targeted degradation [10]. There are approximately 30 million people with a genetic disorder worldwide, and it is estimated that about 30% of them have mutations with NMD-mediated differential allele expression [11]. Importantly, our targeted double-stranded cDNA sequencing-based phase analysis approach also provides information on allele-specific expression and may help for clinical diagnosis and provide a better understanding of the underlying molecular pathology.

Recently, third-generation sequencer such as Oxford Nanopore Technologies (ONT) and Pacific Biosciences (PacBio) can be facilitated for phase analysis due to the production of long sequencing reads (>10 kb) [12,13,14]. The error rates of ONT and PacBio are relatively higher than Illumina NGS [15,16]. The long-read phase test software such as HapCUT and WhatsHap are required for read number and large number of reads including mutations for calculations using statistical algorithms [5,17]. Because these software calculate based on diploid genome, they are not suitable for mRNA; therefore, these software are suitable for comprehensive genome analysis but not for specific genome locus or mRNA. In this study, we verified our targeted double-stranded cDNA sequencing-based phase analysis in a patient with Wilson disease. First, we detected several variants in *ATP7B,* the responsible gene of Wilson disease, using a long PCR-based variant calling method (Figure 1C). Next, we compared the detection ability of compound heterozygous mutations between Nextera tagmentation and ThruPLEX direct adaptor ligation methods for library preparation. We also verified the detection ability of differential allelic expression by targeted double-stranded cDNA sequencing-based phase analysis.

## 2. Materials and Methods

### 2.1. Patient and Sample

A 40-year-old male was suspected of having Wilson disease because of low serum ceruloplasmin value, Kayser–Fleischer ring, and neuropsychiatric symptoms. After genetic counselling, peripheral blood was collected from the patient, and *ATP7B* testing was performed for a definitive diagnosis. Peripheral blood was also collected from the parents of the patient to confirm the compound heterozygous mutations.

### 2.2. Genomic DNA Extraction

We extracted all genomic DNA samples used in this study from peripheral whole blood using a rapid extraction method [18]. The DNA amount and optical density (A260/280 ratio) were measured using a Nanodrop (Thermo Fisher Scientific, Waltham, MA, USA).

### 2.3. Very Long Amplicon Sequencing (vLAS)

Long-range PCR-based NGS, also known as vary long amplicon sequencing (vLAS), was performed at the *ATP7B* genomic region, as previously described [19]. Briefly, a set of very long-range PCR products (approximately 20 kb each) covering the entire gene locus was produced by KOD One (TOYOBO, Osaka, Japan) touchdown PCR. The long PCR primer sequences used in this study are shown in Appendix A. An NGS library was prepared from purified PCR products using a Nextera Flex DNA kit (Illumina, San Diego, CA, USA), according to the manufacturer’s protocol.

### 2.4. Total RNA Extraction and Full-Length Double-Stranded cDNA Synthesis

We extracted total RNA from peripheral blood mononuclear cells with TRIzol reagent (Thermo Fisher Scientific), according to the manufacturer’s instructions. RNA concentration and purity were measured spectrophotometrically (Nanodrop, Thermo Fisher Scientific). The RNA integrity number was determined using a TapeStation 4200 with High Sensitivity RNA ScreenTape (Agilent Technologies, Santa Clara, CA, USA). Full-length double-stranded cDNA was synthesized from 50 ng of total RNA using a SMART-Seq^®^ HT kit (Takara Bio USA, Mountain View, CA, USA), according to the manufacturer’s standard protocol.

### 2.5. Library Preparation for Targeted Double-Stranded cDNA Based Sequencing

We amplified double-stranded cDNA of the *ATP7B* locus (621 bp) by harboring two pathogenic mutations with two specific primers targeting *ATP7B* exon 4/5 and exon 9 (exon 4/5 forward primer, 5′-acattgagctgacaatcacagg-3′ and exon 9 reverse primer, 5′-gagagacatgagtttagccagg-3′). The PCR product was purified using a PCR purification kit (Roche Diagnostics, Mannheim, Germany), and NGS libraries were prepared using either a Nextera XT DNA Library Prep kit (Illumina) or ThruPLEX^®^ Tag-Seq kit (Takara Bio USA), according to the manufacturer’s respective protocol.

### 2.6. Next Generation Sequencing

The libraries were quantified using the HS Qubit dsDNA assay (Thermo Fisher Scientific) and KAPA Library Quantification kit (KAPA Biosystems, Wilmington, MA, USA). According to the standard Illumina protocol, targeted double-stranded cDNA sequencing-based libraries were sequenced (2 × 250 bp) on an Illumina MiSeq. FASTQ files were generated using bcl2fastq (Illumina).

### 2.7. Data Analysis

The FASTQ files in vLAS were aligned to the reference human genome (hg38) using a Burrows-Wheeler Aligner MEM algorithm (BWA-MEM version 0.7.17-r1188) [20]. Haplotype variants were identified using GATK HaplotypeCaller (version 4.0.6.0) [21]. For analysis and interpretation, we used the following software packages: SAMtools (version 1.9), BEDTools (version v2.27.1), vcftools (version 0.1.16), and Integrative Genomic Viewer (IGV 2.4.13), and analysis approach as described previously [22,23,24,25,26,27]. For variant annotation, we used the following databases: SnpEff (version SnpEff 4.3t), dbSNP (version 151), TOPMED, ClinVar, Human Genetic Variation Database (HGVD), and ToMMo (version 3.5) [28,29,30,31]. For in silico analysis, we used dbNSFP (v3.2) that complies a prediction score from 29 prediction algorithms [32]. The original vLAS data presented in this study are available on request from the corresponding author. The vLAS data are not publicly available due to the personal information protection law in Japanese. The raw data of targeted double-stranded cDNA sequencing-based phase analysis in this study cannot be identified by individual information due to short sequence size and have been deposited in the Sequence Read Archive database of NCBI under the BioProject accession number PRJNA699678.

### 2.8. Phase Analysis

The FASTQ files in SMARTer Nextera and SMARTer ThruPLEX were added to unique molecular identifier (UMI, also known as a molecular barcode) information using UMI-tools and were aligned to the reference human genome (hg38) using HISAT2 (version 2.1.0), as described elsewhere [33,34]. The obtained Sequence Alignment/Map (SAM) format files were converted to the Binary Alignment/Map (BAM) file format using SAMtools. Duplicate reads in BAM files were removed using UMI-tools according to UMI information. The BAM files provide the read name and sequence information. We extracted the read name from BAM files with about 10 base-specific sequences around the mutation site. For one mutation (chr13:51958362;G or G>GG), we extracted the read name from BAM files with ATGGGGGGCG and ATGGGGGGGCG, whereas CCCGTGGACC and CCCGTGGATC were used for the other mutation (chr13:51964900;C or C>T). We detected compound heterozygous mutations by comparing the read name, which is common or specific to each mutation.

### 2.9. CHIPS and Sanger Sequencing

To verify the detected haplotype variants, CEL nuclease-mediated heteroduplex incision with polyacrylamide gel electrophoresis and performed silver staining (CHIPS) analysis and Sanger sequencing were performed and direct DNA sequencing, as described previously [26,35].

## 3. Results

### 3.1. Screening for Pathogenic Mutations of ATP7B

Since not only coding region mutations but also deep intronic mutations and large intragenic deletion are known as *ATP7B* pathogenic mutations, very long amplicon sequencing (vLAS) was used for gene mutation screening [36,37]. First, we performed genetic mutation screening of *ATP7B*, the responsible gene of Wilson disease by very long amplicon sequencing (vLAS). The average depth on the *ATP7B* locus was 645, and we detected 168 single nucleotide variants (SNVs) and insertion/deletions (INDELs) variants by haplotype variant calling. These variants were assigned based on functional class as one high, six moderate, four low, and 157 modifiers by SnpEff database. Of those variants assigned as high, we identified a frameshift mutation; we also identified six missense mutations from the moderate category. The allele frequency of all detected variants is approximately 0.5, indicating that all detected variants are heterozygous (Figure 2A). We performed in silico analysis using dbNSFP database which provides the normalized score based on value in the prediction algorithms (SIFT, Polyphen2-HDIV, Polyphen2-HVAR, LRT, MutationTaster, PROVEAN, VEST3, MetaSVM, MetaLR, CADD, DANN, and fathmm-MKL). The scores of the missense variant chr13:51964900;C>T (NM_000053.4:c.1841G>A r.1841g>a p.Gly614Asp) were higher than 0.7 and higher than other variants, indicating that this variant is predicted to cause damage at the protein level (Figure 2B). ClinVar database provides information that the variant (NM_000053.4:c.2304dup r.2304dup p.Met769HisfsTer26) is pathogenic and the variant (NM_000053.4:c.1841G>A r.1841g>a p.Gly614Asp) is likely pathogenic. According to the ACMG guideline of interpretation of sequence variants, 2015, the variant (NM_000053.4:c.2304dup r.2304dup p.Met769HisfsTer26) fulfilled the criteria of pathogenic, and the variant (NM_000053.4:c.1841G>A r.1841g>a p.Gly614Asp) was annotated likely pathogenic [38]. In addition, our analysis using the Trans-Omics for Precision Medicine (TOPMed) which provides allele worldwide database of allele frequency and Japanese population databases such as HGVD and ToMMo indicated that five of these seven variants are polymorphisms, except for two variants chr13:51958362;G>GG (NM_000053.4:c.2304dup r.2304dup p.Met769HisfsTer26) and chr13:51964900;C>T (NM_000053.4:c.1841G>A r.1841g>a p.Gly614Asp) (Figure 2C). Then, on the basis of on these findings, we focused on these two variants, both of which were validated by CHIPS and Sanger sequencing (Figure 2D,E).

### 3.2. Detection of Compound Heterozygous Mutations by Targeted Double-Stranded cDNA Sequencing-Based Phase Analysis

The genomic distance between chr13:51958362;G>GG and chr13:51964900;C>T is 6.5 kb (Figure 3A), whereas the distance in mRNA is only 464 bp, indicating that it is possible to determine whether compound heterozygous mutation is *in trans* or *in cis* mutation using short read targeted double-stranded cDNA sequencing-based phase analysis. First, we amplified the *ATP7B* specific locus double-stranded cDNA (product size: 621 bp) from full-length double-stranded cDNA. Next, we constructed NGS libraries from *ATP7B*-specific double-stranded cDNA using two different methods, i.e., Nextera and ThruPLEX. Nextera is a conventional approach that performs tagmentation and adapter insertion simultaneously using transposon technology. In contrast, we performed direct adapter ligation on both ends of *ATP7B*-specific double-stranded cDNA with the ThruPLEX approach. To evaluate targeted double-stranded cDNA sequencing-based phase analysis performance of these two approaches, we compared mapping rate, coverage, and detection efficiency of mutation and compound mutation. The ThruPLEX mapping rate was higher than that of Nextera (Figure 3B). Furthermore, the ThruPLEX mapping rate without UMI information was almost the same as that of Nextera, indicating that UMI information improves the ThruPLEX mapping rate. Although the coverage of Nextera was low at both ends, we found that the coverage of ThruPLEX was uniform (Figure 3C). The read number, including that of each variant by Nextera was higher than that found by ThruPLEX, reflecting the observation that the mapping read number by Nextera was higher than that by ThruPLEX (Figure 3D,E). We also found that the allele frequencies at both positions (chr13:51958362 and chr13:51964900) were almost identical between Nextera and ThruPLEX (Figure 3F). We also detected paired-end reads including sequences of both positions in ThruPLEX but not Nextera (Figure 3G). The results from ThruPLEX show that chr13:5198362;GG and chr13:51964900;C (chr13:5198362;G and chr13:51964900;T) are within the same allele, indicating that the compound heterozygous mutation is *in trans* mutation. These results indicate that the targeted double-stranded cDNA sequencing-based phase analysis by ThruPLEX approach is better than Nextera as it detects paired-end reads that include the sequences of both positions.

### 3.3. Validation of in trans Compound Heterozygous Mutation by Trio Analysis

To validate our identification of *in trans* compound heterozygous mutation, we investigated whether the patient’s parents possess the two mutations, using CHIPS and Sanger sequencing (Figure 2D,E and Figure 4A). We found that the father had one of the mutations (chr13:51964900;C>T) and the mother had the other mutation (chr13:51958362;G > GG). Moreover, we obtained the same results using RNA-seq analysis (Figure 4B). These results indicate that the patient inherited both mutations from his parents and has *in trans* compound heterozygous mutation (Figure 4C).

### 3.4. Frameshift Mutation Causes Differential Allelic Expression

We also examined whether the two heterozygous mutations cause differential allelic expression. The results obtained from targeted double-stranded cDNA sequencing-based phase analysis indicated that expression of the mutated allele (chr13:51958362;G>GG) was lower than that of the wild type allele (Figure 3F). Furthermore, the expression of the mutated allele (chr13:51958362;G>GG) in the mother was also lower than the wild type allele (Figure 4B). In contrast, expression of the mutated allele (chr13:51964900;C>T) showed the same expression level as the wild type allele. These results suggest that the mutation (chr13:51958362;G>GG) causes differential allelic expression. Therefore, it is possible to detect differential allelic expression, as well as compound heterozygous mutations simultaneously, by targeted double-stranded cDNA sequencing-based phase analysis.

## 4. Discussion

NGS-based applications for clinical diagnosis have advanced and are widely used in Mendelian-inherited diseases and cancer. The intended usage of NGS-based applications including long-read sequencing largely facilitates genome-wide comprehensive analysis such as whole-genome sequencing, whole-exome sequencing, and gene targeting panel sequencing. Several studies have reported phase analysis for genome-wide comprehensive analysis [4,5,6,12]. However, NGS-based phase analysis is rarely reported for a single gene despite often being required for clinical diagnosis.

Our targeted double-stranded cDNA sequencing-based phasing method successfully demonstrated detection of *in trans* compound heterozygous mutations (chr13:51958362;G >GG and chr13:51964900;C>T) in *ATP7B*. This approach has five advantages. First, it overcomes distance limitations by using mRNA instead of genomic DNA. In fact, the average length of mRNA is approximately 2.7 kb as compared with approximately 55 kb for genes. For this reason, it is possible to detect compound heterozygous mutations using mRNA more readily. The distance between our identified compound heterozygous mutations in the *ATP7B* gene tested in this study is 464 bp in mRNA and 6.5 kb in genomic DNA. The second advantage is NGS library size. Although an RNA-seq library size is usually around 300 bp because of fragmentation, it can be extended to up to 1 kb (Illumina NGS library limitation) using our direct adapter ligation method. Thus, it is possible to detect almost any compound heterozygous mutation in our method as the distance limitation of library size is 1 kb as compared with an average mRNA length of 2.7 kb, as long as there is gene expression in peripheral blood. More recently, the PacBio HiFi sequencing method for long-read sequencing was developed [39]. The method yields highly accurate long-read sequencing which provides applications such as single nucleotide and structural variant detection. In addition, it is possible to improve distance limitation in our method by combination with PacBio HiFi sequencing method. Although *ATP7B* expression in blood is low, RT-PCR was still possible as SMARTer cDNA synthesis amplifies full-length cDNA. Importantly, our method detected compound heterozygous mutations in paired-end sequencing reads, which was in contrast to the Nextera method. The third advantage is simplicity. Our approach does not involve new techniques or computationally intense statistical analysis. In fact, our approach has the following three steps: (1) full-length double-stranded cDNA synthesis, (2) targeted amplification and, (3) direct unique molecular identifier (UMI) adapter ligation in library preparation. In addition, the kits required for the third step (SMARTer, KOD One, and ThruPLEX) are commercially available. Furthermore, it was not difficult to detect a compound heterozygous mutation, because paired-end sequencing reads, which harbor approximately 10 specific sequences around each mutation, were directly extracted without the need for installation of specific software. The fourth advantage is the ability to analyze differential allele expression. It has been reported that 9% to 30% of disease-causing mutations have an impact on RNA expression [40]. Therefore, the measurement of mutant allele expression and provision of expression data would confer critically needed information not otherwise readily available for clinical diagnosis. Loss-of-function mutations with the premature stop codon have been identified to cause differential allele expression patterns by NMD [41]. Our findings, in the current study, suggest that the frameshift mutation (chr13:51958362;G>GG) may cause differential allelic expression by decreasing expression of the same allele through NMD. In addition, analyzing differential allelic expression by conventional phase analysis using sanger sequencing is difficult because sanger sequencing of many clones is required. For example, sanger sequencing of at least 1000 clones must be performed for the same level as our targeted double-stranded cDNA sequencing-based phase analysis. Furthermore, differential allele expression analysis cannot be affected by allelic different amplification bias, recombination, and duplication during PCR by UMI technology in targeted double-stranded cDNA sequencing-based phase analysis. Fifth, our method does not require trio analysis. Trio analysis can also detect compound heterozygous mutations; however, it inherently requires sample collection of the patient and both parents, a requirement often difficult to meet in actual clinical practice. In contrast, our method only requires a total RNA sample of the patient only.

A disadvantage of our targeted double-stranded cDNA sequencing-based phase analysis is that intron variants cannot be investigated due to dealing with mRNA. It is required for expression analysis to decide the pathological significance of putative intron variants because most intron variants affect expression and splicing machinery. In practice, targeted double-stranded cDNA sequencing-based phasing needs, for vLAS variant calling, to get information of putative compound heterozygous mutations due to targeted sequencing (Figure 1), therefore, intron variants can be detected during this preliminary step (vLAS). Moreover, targeted double-stranded cDNA sequencing-based phase analysis can investigate differential allelic expression if there are appropriate heterozygous mutations. It might also be possible to analyze low quality fragmented RNA such as FFPE sample because targeted double-stranded cDNA sequencing-based sequencing phase analysis target the short region of double-stranded cDNA.

This study demonstrates that our targeted double-stranded cDNA sequencing-based phase analysis detects compound heterozygous mutations accurately without the need for trio analysis and determines differential allelic expression through NMD by frameshift mutation. We conclude that this method is useful for the determination of compound heterozygous mutations in clinical diagnosis.

## Figures and Tables

**Figure 1 biology-10-00256-f001:**
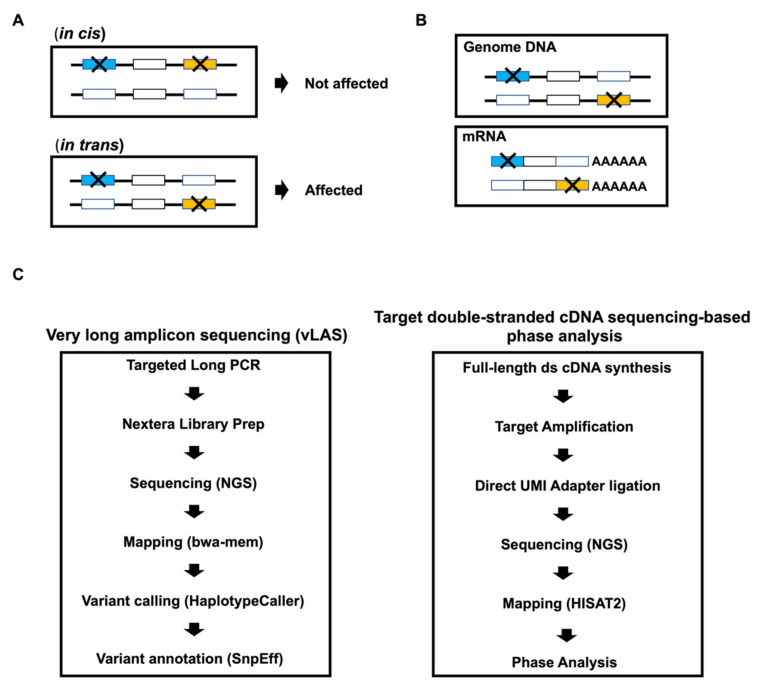
(**A**) Workflow for targeted double-stranded cDNA sequencing-based phase analysis. Types of compound heterozygous mutations (*in cis* and *in trans*); (**B**) Scheme of *in trans* compound heterozygous mutations on genomic DNA and mRNA; (**C**) Workflow for the detection of variants by very long amplicon sequencing (vLAS) and targeted double-stranded cDNA sequencing-based phase analysis.

**Figure 2 biology-10-00256-f002:**
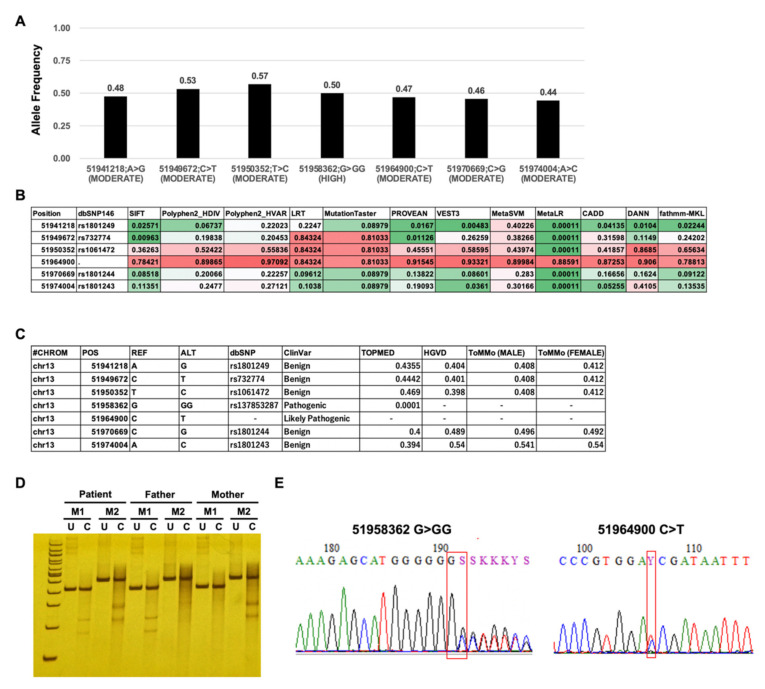
(**A**) Detection of haplotype mutations in the *ATP7B* locus. Allele frequency of variants; (**B**) In silico analysis of variants; (**C**) A summary of variants from the dbSNP, ClinVar, TOPMED, HGVD, and ToMMo databases; (**D**) Trio analysis by CHIPS technology assay (M1: chr13:51964900;C>T, M2: chr13:51958362;G>GG); (**E**) Patient electropherograms of the *ATP7B* mutations loci by Sanger sequencing.

**Figure 3 biology-10-00256-f003:**
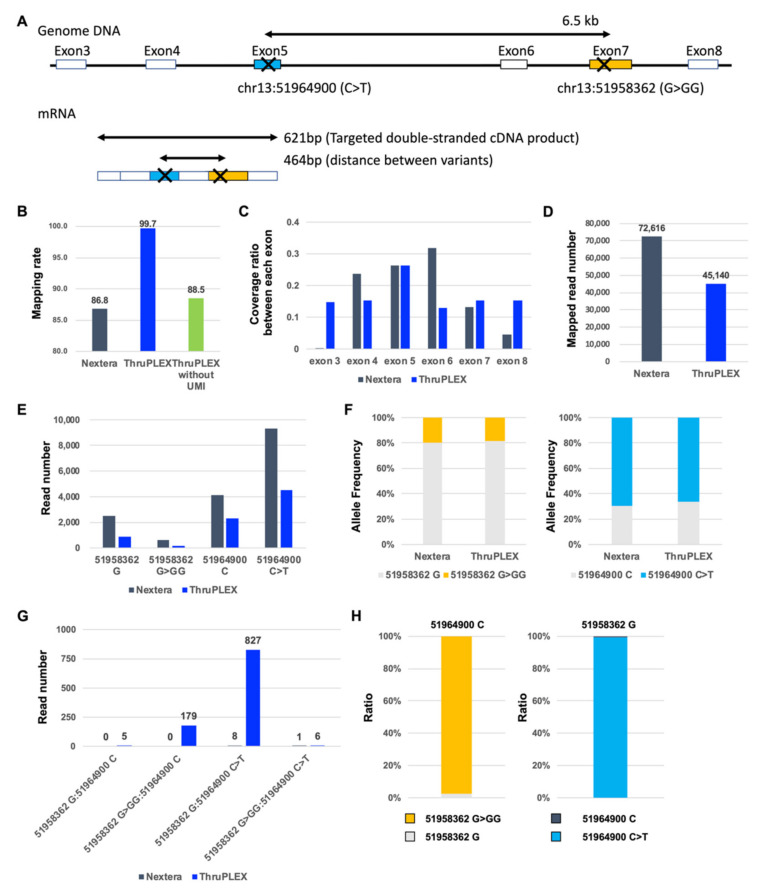
(**A**) Targeted double-stranded cDNA sequencing-based phase analysis. Genomic DNA and mRNA of human *ATP7B*; (**B**) The mapping rate for each sample using different approaches (Nextera versus ThruPLEX); (**C**) The coverage ratio between each exon; (**D**) The mapped read number for each sample; (**E**) The read number harboring each mutation for each sample; (**F**) Allele frequency for each position; (**G**) The read number harboring two mutations for each sample; (**H**) The ratio of reads harboring two mutations.

**Figure 4 biology-10-00256-f004:**
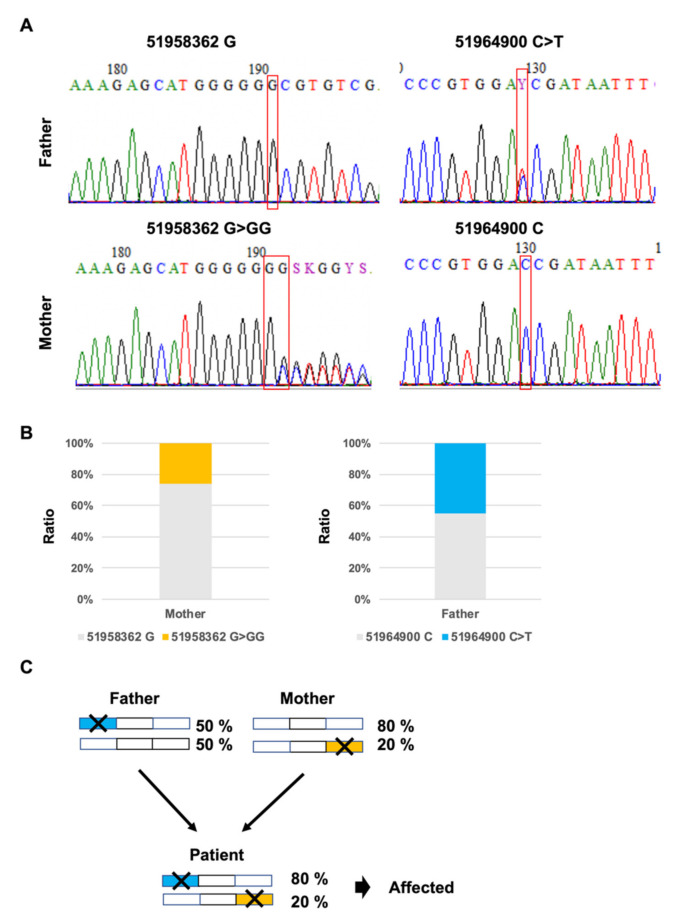
(**A**) Parent electropherograms of the *ATP7B* mutations loci by Sanger sequencing; (**B**) The expressed allele frequency of variants by targeted double-stranded cDNA-based sequencing; (**C**) Predicted inheritance and disease modelling based on targeted double-stranded cDNA-based sequencing.

## Data Availability

The original vLAS data presented in this study are available on request from the corresponding author. The vLAS data are not publicly available due to the personal information protection law in Japanese. The raw data of targeted double-stranded cDNA sequencing-based phase analysis in this study cannot be identified individual information due to short sequence size and have been deposited in the Sequence Read Archive database of NCBI under the BioProject accession number PRJNA699678.

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
