# Peer review of "Targeted Double-Stranded cDNA Sequencing-Based Phase Analysis to Identify Compound Heterozygous Mutations and Differential Allelic Expression"

_biology, 2021, doi:10.3390/biology10040256_

Round 1
Reviewer 1 Report
Ura et al. describe a novel approach for the clarification whether variants are in cis or in trans, which is decisive in recessive diseases. For their approach, the authors make usage of the fact that the distance between two variants on RNA/cDNA level is closer than on gDNA level, which facilitates variant phasing. The authors use two different cDNA library preparation methods and confirm their approach using trio analysis. In addition, the authors show that the RNA-based approach also allows the detection of expression differences of the two alleles, which will help in variant interpretation. Thus, the authors tackle important issues in variant interpretation.
Comments
- Sequence variant description on gDNA and protein levels does not fulfil the HGVS guidelines (https://varnomen.hgvs.org). Likewise, according to HGVS the effect of sequence variants on protein level should be noted as p.(...) if the effect on protein level has not been analyzed on a molecular level (as to my understanding this is the case).
- I would suggest to apply the classification according to the ACMG guidelines instead or in addition to the used classification (page 5 line 189 and Figures 2B and 2C), for which the criteria are missing in the manuscript.
- For long-read sequencing (page 2, lines 87-92) more recent publications should be considered including such on PacBio HiFi reads resulting in lower error rates. Furthermore, long-read sequencing could also be beneficial in the reported targeted cDNA-sequencing approach enabling phasing (i.e. to have both mutations on the same reads).
- It is not completely clear whether only targeted double-stranded cDNA sequencing was performed or also RNA-Seq (cf. e.g. page 4 line 145, page 8 line 254, page 9 lines 259+260).
- On page 4 line 155, the authors mention that they used ClinVar for variant annotation, but I could not find this annotation in Figure 2 as expected.
- TopMed is missing in the methods (page 4 lines 154-157).
- There is conflicting information on the distance of the two sequence variants and the length of the targeted cDNA fragment. Likewise, the stated distance on cDNA level (~400bp on page 1 line 29 and page 10 line 288; 384bp on page 6 line 215 and in Figure 3A) is not consistent with the reported cDNA positions (c.1841G>A and c. 2304dup). Likewise, the length of the targeted cDNA fragment of 621bp can not be reproduced by the provided information. Furthermore, the distance of the two variants on gDNA level is given as 6.5kb (page 1 line 28, page 6 line 214, and Figure 3A) and as 6.7kb (page 10 line 289). Please revise accordingly.
- Based on NM_000053.4, the sequence variant c.1841G>A should be in exon 5 and not in exon 6 as displayed in Figure 3A.
- On page 8 line 252, I assume it should say “Figure 2G” instead of “Figure 2F”.
Author Response
Dear Reviewer,
Thank you for giving us the opportunity to submit a revised draft of our manuscript titled “Targeted double-stranded cDNA sequencing-based phase analysis to identify compound heterozygous mutations and differential allelic expression” to the biology. We appreciate the time and effort that you have dedicated to providing your valuable feedback on our manuscript. We are grateful to you for your insightful comments on our paper. We have been able to incorporate changes to reflect all of your suggestions. We highlighted the changes within the manuscript.
Here is a point-by point response to your comments and concerns.
- Point 1: Sequence variant description on gDNA and protein levels does not fulfil the HGVS guidelines (https://varnomen.hgvs.org). Likewise, according to HGVS the effect of sequence variants on protein level should be noted as p.(...) if the effect on protein level has not been analyzed on a molecular level (as to my understanding this is the case).
Response 1: According to the HGVS guidelines, it said “predicted consequences, i.e. without experimental evidence (no RNA or protein sequence analyzed), should be given in parentheses, e.g. p.(Afg727Ser)”(http://varnomen.hgvs.org/recommendations/protein/variant/substitution/). In this study, we did not put parentheses in the protein sequence because we confirmed the RNA sequence. To emphasize this point, we have made the notation more accurate as follows in Abstract (line 27 and 28 in page 1) and Results (line 201 ~ 214 in page 5); NM_000053.4:c.2304dup r.2304dup p.Met769HisfsTer26 and NM_000053.4:c.1841G>A r.1841g>a p.Gly614Asp.
- Point 2: I would suggest to apply the classification according to the ACMG guidelines instead or in addition to the used classification (page 5 line 189 and Figures 2B and 2C), for which the criteria are missing in the manuscript.
Response 2: Since this classification is from functional class of SnpEff database, we have revised to clarify this point (line 191-195 in page 5). SnpEff and dbNFSP were used for mutation screening, and interpritation were according to ACMG guidelines. We also added the classification according to the ACMG guideline (line 204-210 in page 5).
- Point 3: For long-read sequencing (page 2, lines 87-92) more recent publications should be considered including such on PacBio HiFi reads resulting in lower error rates. Furthermore, long-read sequencing could also be beneficial in the reported targeted cDNA-sequencing approach enabling phasing (i.e. to have both mutations on the same reads).
Response 3: We have added a reference and revised to emphasize this point. Our targeted cDNA-sequencing method still have distance limitation due to short-read sequencing. However, it is possible to improve our method by combination with accurate long-read sequencing method such as PacBio HiFi sequencing method (line 306-310 in page 9).
- Point 4: It is not completely clear whether only targeted double-stranded cDNA sequencing was performed or also RNA-Seq (cf. e.g. page 4 line 145, page 8 line 254, page 9 lines 259+260).
Response 4: We revised to this point. We performed only targeted double-stranded cDNA sequencing (line 146-147 in Page 4 and line 272-274 in page 8).
- Point 5: On page 4 line 155, the authors mention that they used ClinVar for variant annotation, but I could not find this annotation in Figure 2 as expected.
Response 5: We revised to this point and added the variant information of ClinVar in Figure 2C ,line 204-210 in page 5 and line 220 in page 6.
- Point 6: TopMed is missing in the methods (page 4 lines 154-157). There is conflicting information on the distance of the two sequence variants and the length of the targeted cDNA fragment. Likewise, the stated distance on cDNA level (~400bp on page 1 line 29 and page 10 line 288; 384bp on page 6 line 215 and in Figure 3A) is not consistent with the reported cDNA positions (c.1841G>A and c. 2304dup). Likewise, the length of the targeted cDNA fragment of 621bp can not be reproduced by the provided information. Furthermore, the distance of the two variants on gDNA level is given as 6.5kb (page 1 line 28, page 6 line 214, and Figure 3A) and as 6.7kb (page 10 line 289). Please revise accordingly. Based on NM_000053.4, the sequence variant c.1841G>A should be in exon 5 and not in exon 6 as displayed in Figure 3A. On page 8 line 252, I assume it should say “Figure 2G” instead of “Figure 2F”.
Response 6: We revised to this point. We added TOPMED in the methods (line 157 in page 4). The distance on cDNA level is 464 bp, the product size of double-stranded cDNA is 621 bp, and the distance on genomic DNA level is 6.5 kb (line 29 in page 1, line 230 in page 6, line 302-303 in page 9 and Figure 3A). We corrected exon number based on NM_000053.4 (Figure 3A) and correct Figure name (line 265 in Page 8).

Reviewer 2 Report
In the manuscripts the authors describe a method that allows phasing and differential allelic expression based on an targeted double-stranded cDNA sequencing approach. The example of a compound heterozygous variant in ATP7B is used to demonstrate that this method works.
Comments:
- Line 35: ‘caused down-regulation of expression from this allele.’ This can be interpreted in various ways. I assume it is meant that the mutant allele is broken down by nonsense mediated RNA decay due to the frameshift variant, but now it seems that the mutation itself has an effect on the downstream expression of its own allele. Please rephase this.
- It would be good to describe why the vLAS technique is chosen to detect genomic variants.
Furthermore, several (panels of) figures are irrelevant or do not add any value to the manuscript:
- 2A: there is no need to visualize the depth of the target gene, if there is nothing to compare it to à Please remove this part of the figure.
- 2B: There is no explanation in the text where the functional classification annotation into high, moderate, modifier and low come from. Which algorithm is used for it? This functional classification is not relevant to the main message of the manuscript. I suggest to remove panel B as it adds no value at all.
- Also figure 2C seems irrelevant. The point is to show that allelic balance of these variants is approximately 0.5. This information in combination with the respective variant is already shown in panel D. Therefore the information in panel C is redundant and should be removed.
- Figure 2E. The values in this table need to be verified. What kind of CADD-score is used here? A CADD-score of 0.87253 is very low, why is it marked in red? Thresholds for pathogenic CADD scores are usually in the range of >20.
- Figure F. Please provide a dash or NA in the empty cells
Author Response
Dear Reviewer,
Thank you for giving us the opportunity to submit a revised draft of our manuscript titled “Targeted double-stranded cDNA sequencing-based phase analysis to identify compound heterozygous mutations and differential allelic expression” to the biology. We appreciate the time and effort that you have dedicated to providing your valuable feedback on our manuscript. We are grateful to you for your insightful comments on our paper. We have been able to incorporate changes to reflect all of your suggestions. We highlighted the changes within the manuscript.
Here is a point-by point response to your comments and concerns.
- Point 1: Line 35: ‘caused down-regulation of expression from this allele.’ This can be interpreted in various ways. I assume it is meant that the mutant allele is broken down by nonsense mediated RNA decay due to the frameshift variant, but now it seems that the mutation itself has an effect on the downstream expression of its own allele. Please rephase this.
Response 1: We have revised to emphasize this point. It is possible that mRNA harboring the mutant allele was degradated by nonsense mediated decay due to the frameshift variant (line 35 and 36 in page 1).
- Point 2: It would be good to describe why the vLAS technique is chosen to detect genomic variants. Furthermore, several (panels of) figures are irrelevant or do not add any value to the manuscript.
Response 2: We have added references and revised to emphasize this point. Since not only coding region mutations but also deep intronic mutations and large intragenic deletion are known as ATP7B pathogenic mutations, very long amplicon sequencing (vLAS) was used for gene mutation screening (line 187-189 in page 5).
- Point 3: 2A: there is no need to visualize the depth of the target gene, if there is nothing to compare it to à Please remove this part of the figure. 2B: There is no explanation in the text where the functional classification annotation into high, moderate, modifier and low come from. Which algorithm is used for it? This functional classification is not relevant to the main message of the manuscript. I suggest to remove panel B as it adds no value at all. Also figure 2C seems irrelevant. The point is to show that allelic balance of these variants is approximately 0.5. This information in combination with the respective variant is already shown in panel D. Therefore the information in panel C is redundant and should be removed.
Response 3: We have removed unnecessary figures and summarized them as Figure 2A. We used SnpEff database for classification annotation (line 156-157 in page 4 and line 193-194 in page 5).
- Point 4: Figure 2E. The values in this table need to be verified. What kind of CADD-score is used here? A CADD-score of 0.87253 is very low, why is it marked in red? Thresholds for pathogenic CADD scores are usually in the range of >20.
Response 4: We revised to emphasize this point. We performed In silico analysis using dbNSFP database which provide the normalized score of the prediction algorithms including CADD-score (line 198-201 in page 5).
- Point 5: Figure F. Please provide a dash or NA in the empty cells.
Response 5: We provided a dash in the empty cells in Figure 2C.

This manuscript is a resubmission of an earlier submission. The following is a list of the peer review reports and author responses from that submission.
Round 1
Reviewer 1 Report
In their study, Ura et al. describe a novel approach for the clarification whether variants are in cis or in trans, which is decisive in recessive diseases. For their approach, they make usage of the fact that the distance between two variants on RNA level is closer than on DNA level, which facilitates variant phasing. They use two different RNA library preparation methods and confirm their approach using trio analysis. In addition, they show that the RNA-based approach also allows detection of expression differences of the two alleles, which will help in variant interpretation. Thereby, the authors tackle important issues in variant interpretation.
Major comments:
- Although it is implied, the manuscript would benefit from acknowledging long-read sequencing methods (PacBio, Oxford Nanopore), which also facilitate variant phasing by generating reads of several kb in length, as alternative approach.
- Have the authors considered the phasing of variants using dedicated softwares such as WhatsHap (https://whatshap.readthedocs.io/en/latest)?
- Important limitations of the presented approach such as that it is designed for exonic sequence variants only and would need adaption for intronic variants and that RNA, which is not always available, is required are missing in the discussion.
Minor comments
- The genomic position of the deletion is not consistent (cf. chr13:5195862;G>GG (e.g. lines 15, 153, 211, 229, 256) vs. chr13:51958362;G>GG (e.g. lines 179, 192, 214, 229, 240, 241, 244 and Figure 2)).
- On page 7 line 211 for both sequence variants the same genomic position is given: "We also found that the allele frequency at both positions (chr13:5195862 and chr13:5195862)..."
- The manuscript would be easier to read if the disease-causing variants were also described on cDNA and protein level according to the HGVS guidelines (https://varnomen.hgvs.org) and classified according to the ACMG guidelines.
- It is not quite clear why the vLAS data are not publicly available due to privacy reasons, while the RNA sequencing data are available.
- For sequence variant annotation mainly Japanese specific databases are used, but the largest available database gnomAD (https://gnomad.broadinstitute.org) was not considered.
Reviewer 2 Report
In the manuscripts the authors describe a method that allows phasing based on an RNA-sequencing approach. The example of a compound heterozygous variant in ATP7B is used to demonstrate that this method works.
In the example used, the variants are only 400bp away from each other on mRNA/cDNA level. This can easily be investigated using traditional plasmid cDNA cloning followed by sanger sequencing. This traditional method is much cheaper, less labor intensive, faster, and more easily accessible to researchers all over the world. Although it seems the method is capable of detecting the phasing, there seems to be little or no added value of this RNA-seq phasing approach in comparison with traditional cDNA cloning into plasmids combined with sanger sequencing.
Advantages listed in discussion are also applicable to traditional plasmid cDNA cloning + sanger sequencing (e.g. distance advantage, size advantage, differential allele expression, no need for trio analysis). Furthermore argument 3 is not completely valid, as you need some bioinformatics expertise to retrieve the sequences around the mutation. Setting up this pipeline costs a lot more time, compared to traditional methods, and needs to be optimized for every amplicon in every gene.
It seems to be a very expensive approach to confirm whether variants are located in cis or in trans and does not overcome the limitations of traditional approaches. The authors fail to demonstrate the benefits of their approach compared to traditional cDNA cloning + sequencing.
Furthermore, several (panels of) figures are irrelevant or do not add any value to the manuscript (e.g. 2A, 2B, etc.)